# Morals and Reasoning: Formalizing Moral Influence on Reasoning and AI Systems Alignment

Albert Olweny Okiri

Acies Crest

albertokiri@aciescrest.com

June 4, 2025

## Abstract

Reasoning frameworks are built upon ethical principles or the lack thereof. This paper provides a psychoanalytical perspective on reasoning patterns and dynamics, focusing on the impact of moral values on decision-making and social outcomes. We formalize how moral values shape perception and enhance institutional performance, introducing the Values DIffusion Effect to describe their positive, spreading influence. We provide a systematic formalization for implementing ethical evaluation, offering a standard and replicable process for incorporating ethically guided decision-making. We examine how institutions can sustainably evolve by effectively considering dynamic external conditions through efficient decision-making systems and social interactions. For example, in economic institutions, systems and processes can evolve through ethical consideration of external factors to achieve a sustainable state, addressing issues like unbounded economic objectives, which should be tied to market satisfaction and sustainability. In artificial intelligence (AI), we propose technical considerations for AI alignment that build on findings from the sociological domain regarding ethically grounded reasoning. Simulation strategies are also explored, leveraging the ability of autoregressive models to accurately estimate social contexts within generative agent-based models (GABMs) using frameworks such as Concordia to measure efficiency gains from ethically oriented reasoning frameworks. Applications from these findings include sustainable economic institutions and ethically aligned AI systems, addressing challenges like resource allocation and AI decision-making.

## Prologue

*This paper draws from scientific psychosocial insights and examination of social contexts. Consistent patterns unearthed from historical contexts, research studies, religious texts, and their holistic examination provide the concepts expressed herein and reinforce confidence in their accuracy. Although the concepts may be perceived as novel, they represent unexplored interpretations of psychosocial patterns compiled to create a new paradigm for decoding psychological mechanics. The concepts are laid out in simple, logical steps to present the reader with a clear psychoanalytic perspective applicable to any context. This work aims to establish a more accurate foundational framework for psychoanalysis, enhancing understanding of social dynamics in both simple and complex environments. From this clear understanding of the human condition, sustainable social systems and institutions can emerge. Contemporary challenges, such as climate change and inefficiencies in economic systems, can be addressed through widely accepted, efficient systems for evaluating contexts. These systems can be established and propagated by introducing and promoting moral values within social settings.*

## 1   Introduction

The quality of decisions and the sustainability of their outcomes in social settings and institutions depend on the accuracy of projections and the selection of positive, long-term choices from those possibilities. Moral consideration during decision-making provides an effective method for ensuring the realization and discovery

of sustainable outcomes. This standard approach is universally applicable to social considerations and efficient at addressing social challenges and systemic limitations that undermine the performance of current institutions. The evolution of institutions, conceptualized as the effective evaluation of external or internal factors and the implementation of optimal changes, can be guaranteed with robust reasoning frameworks. Inefficiencies in institutions, such as governance systems, highlight the limitations of current social systems and underscore the need for ethical decision-making frameworks [3].

This paper focuses on the evaluation and enactment of decisions through moral values and their impact on social dynamics and institutional performance. We explore how rationale, rooted in moral values, shapes perceptions, behaviors, and political orientations across individuals and groups. By examining the interplay between moral values, social interactions, and institutional outcomes, we aim to formalize a methodology for fostering sustainable institutions that accommodate diverse ideological perspectives while promoting ethical coherence and social efficiency [7]. Through simulations, we select institutions with accurately measurable metrics, such as economic performance, to propose strategies for measuring efficiency within social environments and institutions following the introduction of moral values.

## 2  Psychoanalytical Decomposition and Formalization

We decompose reasoning frameworks within social settings and validate the concept of introducing moral values and ethics as an approach to enhancing decision-making frameworks. By tracing the progression from ethics to political filters and resulting psychosocial perspectives, we establish a system for analyzing social behavior and dynamics tied to resonance with moral values. Political filters, as an abstract concept of cognition, refer to socially accepted standards emergent from sets of moral values that resonate across groups and individuals and are applied to efficiently evaluate social contexts. Notions and popular psychosocial perspectives are popularized and validated through the consistent evaluation of social conditions and ideologies across groups or networks. The mutuality of these social patterns in networks is determined by the resonance of specific political filters, which are formed by the set of moral values they comprise. These dynamics between psychosocial perspectives and social behavior demonstrate the role that moral values and ethics play in rationality, social outcomes, and the sustainability of institutions [9].

Formally, let:

- $M$: The set of moral values in a society, representing principles such as fairness, justice, and empathy.

- $P$: A political filter, defined as a subset of $M$ (i.e., $P \subseteq M$) or the entire set $M$, encapsulating a selection of moral values that guide social evaluations.

- $C$: The set of social contexts, comprising situations or environments where evaluations occur.

The evaluation function is defined as:

$$E : C \rightarrow P, \quad E(c) = P(c),$$

where $E(c)$ assigns a political filter $P(c) \subseteq M$ to a social context $c \in C$. Thus, the evaluation of a social context is equivalent to the application of a political filter, as $E(c) = P(c)$.

The social welfare function measures the quality of outcomes:

$$W : P \rightarrow \mathbb{R},$$

where $W(P(c))$ quantifies the social benefit of applying political filter $P(c)$ to context $c$. Enhancing the moral values within $P$, such as incorporating more inclusive or equitable values (e.g., expanding $P$ to include additional elements from $M$), improves the evaluation. Formally, if $P' \supseteq P$ (i.e., $P'$ includes more or enhanced moral values from $M$), then:

$$W(P'(c)) \geq W(P(c)),$$

indicating that a richer set of moral values in the political filter enhances social outcomes.

To model social dynamics, we examine how moral values influence decisions and the behavior of others, shaping collective outcomes. Moral values guide individual and group decision-making, fostering trust and

cooperation across social networks. Consider a social interaction network where agent $i$'s decision $d_i \in D$ is influenced by moral values $m_i \in M$:

$$d_i = h(m_i, c, \{d_j\}_{j \neq i}),$$

where $h$ is a decision function, $c \in C$ is the context, and $\{d_j\}_{j \neq i}$ represents decisions of other agents. Moral influence propagates through the network, as fairness by agent $i$ may elicit reciprocal fairness, enhancing social cohesion and sustainable outcomes [10, 9].

**Diagram of Moral Influence on Social Outcomes** To illustrate the formalization, Figure 1 shows the relationship between moral values $(M)$, political filters $(P)$, social contexts $(C)$, and social welfare $(W)$. The diagram uses a Venn diagram to depict $P \subseteq M$, with colorful nodes and directed edges representing the evaluation process and its impact on social outcomes.

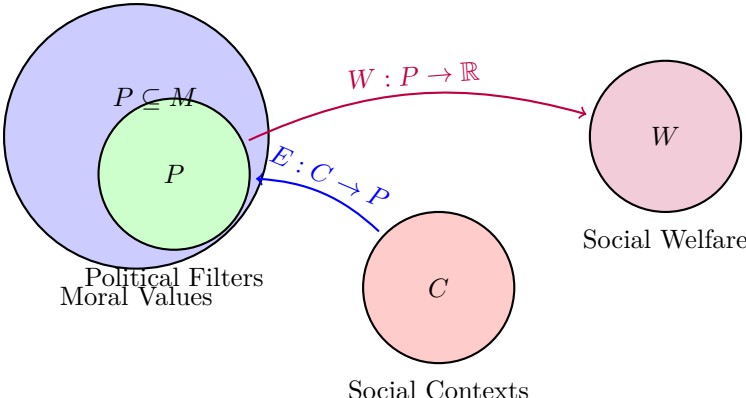

Figure 1: Venn diagram illustrating the flow from moral values $(M)$ to political filters $(P$, a subset of $M)$ through the evaluation function $(E)$ applied to social contexts $(C)$, leading to social welfare outcomes $(W)$.

## 3 Systematic Design for Ethical Evaluation

To ensure ethical and sustainable decision-making, we propose a systematic approach to evaluating contexts, focusing on isolating key facets within a context and assessing their sustainability over time. Facets are distinct components or elements within a context (e.g., economic indicators, social norms, environmental factors) that influence outcomes. By computing the sustainability of each facet—defined as its likelihood of exhibiting positive changes over time—decision-makers can prioritize actions that enhance long-term social welfare. This evaluation process considers three key factors: (1) the objective or task, (2) the action to be enacted, and (3) the evolution or change over time. These considerations can be applied cumulatively, individually, or in a specific order, depending on the context [8].

Formally, let:

- $C$: A context, comprising a set of facets $F = \{f_1, f_2, \ldots, f_n\}$.

- $S(f_i, t)$: The sustainability score of facet $f_i \in F$ at time $t$, representing the likelihood of positive change.

- $O$: The objective or task (e.g., asset allocation in economic contexts).

- $A$: The set of possible actions $\{a_1, a_2, \ldots, a_m\}$.

- $E(f_i, a, t)$: The evolution function, estimating the change in facet $f_i$ under action $a \in A$ over time $t$.

The ethical evaluation function for a context is:

$$V : C \times A \to \mathbb{R}, \quad V(c, a) = \sum_{f_i \in F} w_i \cdot S(f_i, t_a),$$

where $w_i$ is the weight of facet $f_i$ (reflecting its importance in the context), and $t_a$ is the time horizon associated with action $a$. The sustainability score $S(f_i, t_a)$ is computed as:

$$S(f_i, t_a) = g(O, a, E(f_i, a, t_a)),$$

where $g$ aggregates the objective, action, and temporal evolution to estimate sustainability. The optimal action $a^*$ is:

$$a^* = \arg\max_{a \in A} V(c, a),$$

prioritizing actions that maximize weighted sustainability across facets.

This approach requires understanding the composition of facets, their relationships with the environment, and changes that enhance their quality or optimize their processes. Facets with the highest positive influence—those contributing most to sustainable outcomes—are prioritized. The considerations (objective, action, evolution) can be applied flexibly. For example, the objective might guide the initial selection of actions, while temporal evolution refines the assessment of sustainability.

**Example: Asset Allocation in Economic Contexts** Consider an economic context where the objective is asset allocation to optimize market performance while ensuring sustainability. The context includes facets such as market stability ($f_1$), environmental impact ($f_2$), and social equity ($f_3$). Possible actions include allocating funds to renewable energy ($a_1$), fossil fuels ($a_2$), or technology startups ($a_3$). The sustainability score for each facet is estimated based on the objective (maximizing returns), the action (specific allocation), and temporal evolution (projected changes over 10 years). For instance: - Action $a_1$ (renewable energy) may yield high $S(f_2, t)$ due to reduced environmental impact and moderate $S(f_1, t)$ due to stable returns. - Action $a_2$ (fossil fuels) may have high short-term $S(f_1, t)$ but low $S(f_2, t)$ due to environmental degradation. - Action $a_3$ (tech startups) may balance $S(f_1, t)$ and $S(f_3, t)$ by fostering innovation and job creation.

The evaluation function $V(c, a)$ computes the weighted sum of sustainability scores, with weights reflecting priorities (e.g., $w_2$ higher for environmental impact). Action $a_1$ may be selected if it maximizes $V(c, a)$, reflecting its high sustainability across facets. This example illustrates how the systematic approach prioritizes actions with the greatest positive influence, requiring an understanding of facet interactions and their long-term effects [12].

# 4 AI Alignment

In artificial intelligence (AI), ensuring that systems make decisions aligned with universally accepted ethical principles is critical, particularly as their deployment expands across domains such as healthcare, finance, and governance. Traditional alignment methods, such as reward shaping, imitation learning, or value learning, often fall short in fully automated settings due to challenges in scalability, generalization, and handling complex ethical dilemmas. These methods struggle to adapt to the nuanced and context-dependent nature of ethical decision-making, especially in scenarios where AI systems must operate autonomously without human intervention.

A promising approach to addressing these challenges is the Joint-Embedding Predictive Architecture (JEPA), proposed by Yann LeCun in his influential paper *A Path Towards Autonomous Machine Intelligence* [5]. JEPA represents a shift toward building AI systems that learn, reason, and plan more like humans by incorporating world models and cost modules. These components are designed to evaluate the ethical implications of actions within a given context, ensuring that decisions are not only efficient but also aligned with human values such as fairness, equity, and sustainability.

Broad generalization of implementing standard reasoning for AI using ethics can be achieved by drawing from ethical influences in reasoning and efficiency gains observed in social institutions. For instance, AI systems can be designed to consider all facets within an environment or context sustainably, ensuring alignment with ethical principles. This involves reasoning about the world based on the task or objective and breaking down the facets that need to be evaluated for sustainability based on the action or projected evolution over time. By choosing optimal strategies or actions based on least-cost pathways that are also the most ethical, AI systems can achieve both efficiency and alignment.

Perception models for ethical reasoning in AI systems can be further enhanced by integrating world models and cost modules, as suggested in the JEPA framework. These models allow AI to simulate the

consequences of actions across time and evaluate their ethical impact on various facets of a context. For example, in a decision-making scenario involving resource allocation in healthcare, an AI system could assess the sustainability of different actions by considering their long-term effects on patient outcomes, cost-effectiveness, and equitable access. The evaluation function can be formalized as follows:

Let:

- $C$: A context, comprising a set of facets $F = \{f_1, f_2, \ldots, f_n\}$, such as patient health, resource availability, and equity.

- $S(f_i, t)$: The sustainability score of facet $f_i \in F$ at time $t$, representing the likelihood of positive change.

- $O$: The objective or task (e.g., optimizing healthcare delivery).

- $A$: The set of possible actions $\{a_1, a_2, \ldots, a_m\}$, such as allocating funds to different treatments.

- $E(f_i, a, t)$: The evolution function, estimating the change in facet $f_i$ under action $a \in A$ over time $t$.

The ethical evaluation function for a context is:

$$V : C \times A \to \mathbb{R}, \quad V(c, a) = \sum_{f_i \in F} w_i \cdot S(f_i, t_a),$$

where $w_i$ is the weight of facet $f_i$, reflecting its importance, and $t_a$ is the time horizon associated with action $a$. The sustainability score $S(f_i, t_a)$ is computed as:

$$S(f_i, t_a) = g(O, a, E(f_i, a, t_a)),$$

where $g$ aggregates the objective, action, and temporal evolution to estimate sustainability. The optimal action $a^*$ is:

$$a^* = \arg\max_{a \in A} V(c, a),$$

prioritizing actions that maximize weighted sustainability across facets.

This approach ensures that AI systems prioritize actions with the greatest positive influence, requiring an understanding of facet interactions and their long-term effects. For instance, in healthcare, allocating resources to preventive care may yield higher sustainability scores for patient health and equity compared to short-term treatments, aligning with ethical goals.

In summary, AI alignment can be significantly advanced by adopting architectures like JEPA, which incorporate ethical evaluation through world models and cost modules. This approach enables AI systems to reason ethically, generalize across contexts, and make decisions that are both efficient and aligned with human values, addressing challenges in scalability and ethical complexity.

# 5  Ethical Influence on Social Network Dynamics

The influence of moral values on social network dynamics is profound, shaping both individual behaviors and collective outcomes in physical and digital social environments. Ethics play a crucial role in fostering sustainable social interactions by introducing care and consideration for all facets within a context, leading to sustainable representations and perceptions. This influence manifests in several key ways, categorized as follows:

1. **Introducing Care and Consideration of Facets Within a Context:**

   - Ethical frameworks encourage the provision of information that considers all facets of a context sustainably. This includes emotional regulation, positive emotional shifts, and curation of perceptions that are considerate of the entirety of facets within a context, such as individual rights, community well-being, and environmental impact.

- Such approaches lead to more accurate and efficient resolution of contextual issues, fostering truth and factual understanding. For example, in online social networks, ethical guidelines can promote respectful dialogue and prevent the spread of misinformation, ensuring that discussions are constructive and beneficial to all parties involved [2].

2. **Identification and Elimination of Unsustainable Stances/Percepts:**

- Ethics provide a systematic approach to identifying behavioral aspects during social discourse that lead to unsustainable outcomes, such as immoral concepts or harmful ideologies. For instance, behaviors that promote division or misinformation can undermine social cohesion.

- By expressing alternative viewpoints that are considerate of all parties' interests and contexts, ethics help eliminate these unsustainable percepts. This process not only resolves conflicts but also strengthens social cohesion by promoting fairness and inclusivity.

In the context of social media and online interactions, ethical considerations are vital for maintaining respectful and constructive dialogue. Organizations like the American Speech-Language-Hearing Association (ASHA) emphasize the ethical use of social media, urging users to avoid personal attacks and engage in discussions that are beneficial to the community [2]. Similarly, ethical guidelines for social network analysis (SNA) highlight the importance of respecting privacy and ensuring that no harm is caused to individuals through the analysis of their social connections [6]. Key ethical concerns in SNA include violation of privacy, psychological harm, and harm to individual standing, which must be carefully managed to maintain trust.

Moreover, ethics influence social network dynamics by addressing the privacy paradox, where users share personal information despite privacy concerns, leading to potential ethical issues such as data misuse or security breaches [4]. Ethical frameworks can help mitigate these issues by promoting transparency, respect, and responsibility in how data is collected, shared, and used. For example, ethical social media use involves being transparent about affiliations and ensuring content is accurate and not misleading [1].

The impact of ethical considerations can be formalized as follows:

Let:

- $N$: A social network, represented as a graph $N = (V, E)$, where $V$ is the set of nodes (individuals or entities) and $E$ is the set of edges (relationships).

- $M$: The set of moral values, such as fairness, respect, and transparency.

- $B_i$: The behavior of node $i \in V$, influenced by moral values $m_i \in M$.

- $S(N, t)$: The sustainability score of the network at time $t$, reflecting the quality of interactions and outcomes.

The behavior of a node is modeled as:

$$B_i = h(m_i, c, \{B_j\}_{j \neq i}),$$

where $h$ is a function that maps moral values, context $c$, and the behaviors of other nodes to the behavior of node $i$. The sustainability score of the network is:

$$S(N, t) = \sum_{i \in V} w_i \cdot Q(B_i),$$

where $Q(B_i)$ measures the quality of node $i$'s behavior (e.g., degree of respectfulness or fairness), and $w_i$ is the weight of node $i$'s influence in the network. Ethical behaviors, guided by $M$, increase $S(N, t)$, leading to more sustainable and cohesive network dynamics.

By integrating ethical principles into social network dynamics, we can create environments that promote sustainability, fairness, and positive social outcomes, addressing challenges like misinformation, privacy breaches, and social division.

| Network Indicator | Description |
|---|---|
| Interaction Quality | Degree of respectfulness and constructiveness in communications. |
| Information Accuracy | Prevalence of truthful and non-misleading content. |
| Privacy Respect | Adherence to ethical guidelines protecting user data. |
| Conflict Resolution | Efficiency and fairness in resolving disputes. |
| Social Cohesion | Strength of trust and cooperation among nodes. |

Table 1: Indicators of Ethical Influence on Social Network Dynamics

# 6 Decision-making and Sustainable Institutions

The performance of institutions, beyond their ability to chart sustainable paths, relies on the effectiveness of social interactions, which influences their optimal operational capability. Bureaucracy and the efficiency of enacting decisions directly impact institutional performance, tied to the quality of social interactions. Identifying sustainable pathways requires a decision-making process that ethically evaluates factors within the institutional context, yielding actions and perceptions that ensure long-term sustainability. The evolution and adaptation of institutions depend on efficiency gains from embedding moral values into social settings. Distorted or incomplete moral values in political filters can lead to reasoning frameworks that promote unsustainable outcomes, creating a cascading effect that negatively impacts institutions and processes [3, 8].

Formally, let $D$ be a decision-making process, defined as:

$$D : I \rightarrow A,$$

where $I$ represents the institutional context, encompassing internal and external factors as percepts (e.g., social norms, economic constraints, environmental conditions), and $A$ is the set of sustainable actions or perceptions. These factors are ethically evaluated to produce actions that promote long-term sustainability. The quality of decisions is evaluated by:

$$Q(D) = W(A),$$

where $W$ is the social welfare function. Ethical decisions maximize $Q(D)$ by aligning actions with moral values $M$, ensuring sustainable and socially beneficial institutional outcomes [10].

## 6.1 Economic Institutions

Moral considerations in decision-making not only improve the quality of processes within a context but also shape economic incentives. Ethical reasoning frameworks promote sustainable outcomes, leading to effective resource placement in financial markets. Efficient markets dynamically allocate resources based on available information, with ethical considerations ensuring alignment with social and environmental goals. This approach addresses challenges like climate change and inefficient resource utilization by fostering sustainable economic models. Policymakers can design regulations that ethically consider external factors, leading to economic systems that prioritize long-term sustainability [12, **?**].

Formally, let $R$ be resources and $A_R$ be allocation actions. The allocation function is:

$$A_R : R \times M \rightarrow P,$$

where $P$ represents the determination of resource placement, and $M$ is the set of moral values (e.g., fairness, equity, sustainability). Ethical considerations ensure:

$$W(A_R(r, m)) > W(A_R(r, \emptyset)),$$

where $m \in M$ and $\emptyset$ denotes no moral input, indicating that morally informed resource placement enhances social welfare and sustainability.

# 7    Simulations Strategy

Capturing the nuances of social interactions in simulations is critical for creating a credible empirical design to ascertain efficiency gains from introducing moral values in social and economic institutions. The complexity and abstract nature of social dynamics highlight the limitations of traditional agent-based designs for simulating social settings. The advent of large language models provides a unique opportunity to represent abstract social contexts using natural language instructions and extrapolate these conditions to next-state outcomes, mimicking real-life social interaction scenarios. Generative agent-based models (GABMs) leverage large language models to represent social contexts during simulations.

Introducing additional social contexts, such as economic incentives, provides well-known and measurable aspects to understand the efficiency of social institutions and the impact of moral values. By examining processes and aspects such as economic performance and social network patterns, we create a robust empirical approach to understanding the role of moral values.

Using the Concordia framework, we simulate how moral values influence decision-making and social outcomes in economic institutions. By varying agents' moral adherence parameters ($\alpha_i$), we measure efficiency indicators such as resource utilization, cooperation levels, and conflict resolution. Economic institutions provide well-established metrics to quantify the effectiveness of social interactions under different ethical frameworks. The methodology involves:

- **Accurate Simulation of Nuances**: Agents reflect diverse moral perspectives, capturing subtle variations in social behavior.

- **Documentation of Influences**: The Game Master logs interactions and outcomes, providing a detailed dataset for analyzing moral impacts.

- **Metric-Driven Analysis**: Metrics like resource allocation efficiency and agent satisfaction evaluate the role of moral values in institutional performance.

**Concordia Framework Operation** The Concordia framework, developed by Google DeepMind, enables the simulation of complex social interactions through GABMs [11]. The framework consists of agents (nodes) interacting via natural language actions, mediated by a Game Master (central controller). Edges represent interactions, and the Game Master ensures environmental consistency. This structure allows researchers to study how moral values propagate through social networks, amplifying cooperation and efficiency in simulated economic settings.

| Indicator | Description |
|---|---|
| Resource Utilization | Effectiveness of resource management or sharing. |
| Task Completion | Speed and quality of task completion. |
| Cooperation Levels | Frequency and quality of cooperative interactions. |
| Conflict Resolution | Efficiency and peacefulness of conflict resolution. |
| Agent Well-being | Overall satisfaction or happiness of agents. |
| Emergence of Social Structures | Formation of norms or institutions promoting efficiency. |

Table 2: Efficiency Indicators for GABM Simulations

By simulating economic institutions with Concordia, we demonstrate improved cooperation, resource placement, and social efficiency with higher moral adherence, reflecting the impact of moral values on institutional sustainability.

# 8    Conclusion

This paper formalizes the role of moral values in reasoning and institutional sustainability, using psycho-analytical, social-dynamic, and AI-based models. The Values Diffusion Effect illustrates how moral values amplify social cohesion and sustainable performance. Simulations with the Concordia framework validate

efficiency gains from ethical considerations in social interactions and institutional outcomes. The proposed frameworks, including systematic ethical evaluation and AI alignment with world models and cost modules, offer actionable methodologies for designing sustainable economic systems and ethically aligned AI. Future work includes refining simulation methodologies to capture finer nuances of social dynamics and exploring moral impacts in AI and biological systems [3, 7].

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
