# OpenReview forum: "Morals and Reasoning: Formalizing Moral Influence on Reasoning and AI Systems Alignment"
_colmweb.org/COLM/2025/Workshop/Social_Sim — Social Sim'25_

### Official Review · Reviewer_YCom · 2025-07-14
**A psychoanalytical and formal framework for moral reasoning**

**Rating:** 6
**Overall Assessment:** 3
**Confidence:** 4

**Review:**

The paper presents an original interdisciplinary framework for moral reasoning with formal modeling and simulation strategies. The proposed formalism generalizes to multiple domains such as economics and AI. The pros involve potential application for multi-agent LLM social simulations, which is highly relevant for enhancing AI safety and reasoning abilities. However, the writing is rather philosophical and theoretical, lacking technical groundedness and real-world case studies. Further, it lacks comparison with existing literature in the domain. Overall, the paper might be of some interest to non-technical policymakers in the sociotechnical track at the workshop.

**Comments Suggestions And Typos:**

* Typo on Page 7, Section 6.1: Missing reference after “prioritize long-term sustainability [12, …]”
* Including real-world examples or case studies would help to understand the practicality of the proposed formalism.

**Paper Summary:**

This paper provides a psychoanalytical and formal approach to understanding how moral values shape reasoning processes, influence institutional sustainability, and guide AI alignment. The author introduces the Values Diffusion Effect as a formal framework to quantify the influence of moral values on decision-making and social outcomes, incorporating social context, political filters, social welfare, moral facets and their temporal sustainability scores. The paper presents an ethical evaluation function for a given context using a weighted sum of moral facet sustainabilities and proposes picking the optimal action that maximizes this function. The paper applies these ideas to institutional performance, AI alignment via JEPA-inspired architectures, and simulations using generative agent-based models like Concordia. Overall, the work calls for the systemic embedding of moral reasoning to improve both human and AI decision-making efficacy and sustainability.

**Relevance:**

4

**Summary Of Strengths:**

* The paper proposes a unified moral reasoning framework across diverse domains, such as economics and AI, formally defining ethical evaluation functions and sustainability scoring.
* The proposed Concordia + GABM simulation framework with JEPA inspiration is highly relevant to multi-agent social simulations and moral reasoning for LLMs.

**Summary Of Weaknesses:**

* Most of the formalism presented in the paper is highly theoretical and simulation-driven, with no real-world experiments or case studies.
* The Concordia experiments lack quantitative analysis and results – only high-level qualitative findings are presented.
* The paper lacks a dedicated Related Work section to understand where the proposed approach fits in with existing literature.

---

### Official Review · Reviewer_toeP · 2025-07-16

**Rating:** 3
**Overall Assessment:** 2
**Confidence:** 4

**Review:**

The exact research question of the paper is unclear. Although the author proposes many theories, without any experiments, it is hard to determine their utility. For example, in Section 7, the author discusses conducting simulations using Concordia but does not share any results or further details of the experiments. The work seems incomplete.

**Comments Suggestions And Typos:**

N/A

**Paper Summary:**

The paper formalizes moral consideration during decision making. Incorporating several factors such as facets, contexts, objectives, etc, in consideration, they provide a framework for evaluating moral values and ethics for decision making. The paper argues why traditional alignment methods fall short in AI and theoratically shows how using their formulation can offer a better alternative, akin to the JEPA framework. They further motivate the theory for ethical considerations in social networks and economic institutions. Finally, they use the Concordia framework to simulate how moral values influence decision-making and social outcomes in economic institutions.

**Relevance:**

2

**Summary Of Strengths:**

The proposed theory could be beneficial only if it is well-motivated with some evidence.

**Summary Of Weaknesses:**

See Review

---

### Meta-Review · Program_Chairs · 2025-07-24

**Recommendation:** Accept

**Metareview:**

--